# Prediction of the Long-Term Performance and Durability of GFRP Bars under the Combined Effect of a Sustained Load and Severe Environments

**DOI:** 10.3390/ma13102341

**Published:** 2020-05-19

**Authors:** Jianwei Tu, Hua Xie, Kui Gao

**Affiliations:** 1State Key Laboratory of Silicate Materials for Architecture, Wuhan University of Technology, 430070 Wuhan, China; tujianwei@whut.edu.cn; 2Hubei Key Laboratory of Roadway Bridge and Structure Engineering, Wuhan University of Technology, 430070 Wuhan, China; xiehua_tujian@163.com; 3General Constraction Company, China Construction Third Engineering Bureau Co., Ltd., 430070 Wuhan, China

**Keywords:** GFRP bars, durability, stress level, degradation, long-term performance prediction

## Abstract

With the continuous development of production technology, the performance of glass-fiber-reinforced polymer (GFRP) bars is also changing, and some design codes are no longer applicable to new materials based on previous research results. In this study, a series of durability tests were carried out on a new generation of GFRP bars in laboratory-simulated seawater and a concrete environment under different temperatures and sustained loads. The durability performance of GFRP bars was investigated by analysing the residual tensile properties. The degradation mechanism of GFRP bars was also analysed by scanning electronic microscopy (SEM). Furthermore, the long-term performance of GFRP bars exposed to concrete pore solution under different stress levels was predicted using Arrhenius theory. The research results show that the degradation rate of GFRP bars was increased significantly at a 40% stress level. By comparing the test results, design limits, and other scholars’ research results, it is demonstrated that the GFRP bars used in this test have a good durability performance. It is found that the main degradation mechanism of the GFRP bars is the debonding at the fiber-matrix interface. In the range test, the effects of a 20% stress level on the degradation of GFRP bars were not obvious. However, the long-term performance prediction results show that when the exposure time was long enough, the degradation processes were accelerated by a 20% stress level.

## 1. Introduction

The steel bars in traditional reinforced concrete (RC) structures are subject to increasing corrosion due to the continuous intrusion of chloride ions in seawater, which reduces the durability of the RC structures. Costly remedial measures, such as coating the surface of the steel bar with epoxy or galvanized steel using stainless steel bars, have failed to improve the long-term durability performance of RC structures [1]. In recent years, fiber-reinforced polymer (FRP) bars have been increasingly used in civil engineering and are expected to serve as an ideal alternative to steel bars because of their resistance to corrosion. Due to the lower cost of glass-FRP (GFRP) bars, they are more widely used than aramid-FRP (AFRP) and carbon-FRP (GFRP) bars. However, glass fibers and commonly used thermosetting resin matrices are known to be susceptible to erosion by moisture, a high temperature, and alkali conditions. GFRP bars may undergo polymer degradation, fiber-matrix debonding, and fiber corrosion, which can cause a significant reduction in the tensile strength [2,3]. Therefore, studying the durability of GFRP bars is very important for the design of GFRP-RC structures.

The degradation rate of GFRP bars depends on the rate of the corrosion ions entering the interior of the GFRP bars and the rate of the corrosion ions reacting with the matrix and fibers. Therefore, the durability of GFRP bars tends to be affected by many factors, such as 1) environmental factors, such as the pH [3,4], exposure temperature [5], moisture content in concrete [6,7], and sustained stress level [8,9], and 2) production process factors, such as the bar diameter [9,10,11], fiber type [8], matrix type [12,13], cross-sectional shape [14], fiber-matrix bonding process [13,14], and so on. With the continuous development of production technology, the performance of GFRP bars is also changing, and some design codes are no longer applicable to new materials based on previous research results. Therefore, it is necessary to study the durability of new GFRP bars and compare the results with the literature, so as to revise the design codes accordingly. At the same time, the research data can help manufacturers produce more economical GFRP bars.

In GFRP-RC structures, GFRP bars normally work under stress. Stress may cause micro-cracks in the matrix. These micro-cracks may result in the invasion of the surrounding environment (such as hydroxyl ions (OH–) and moisture, etc.), which in turn may corrode the fibers [15,16]. According to the research results of Benmokrane et al. [8], different stress levels may cause three types of degradation mechanisms for GFRP bars: 1) at low stress levels, defects in the matrix are not sufficient to form micro-cracks, and the fibers are not exposed to the corrosive environment. Corrosive mediums can only enter by diffusion; 2) at moderate stress levels, defects in the resin begin to form micro-cracks, and the transmission rate of the corrosive medium entering the interior of the GFRP bar is greatly increased, significantly reducing the service life of the GFRP bar; 3) under high stress levels, the internal cracks in the matrix continue to increase, and excessively high stress levels may even cause the fibers to break, leading to a sharp reduction in the service life of the GFRP bar. Benmokraneet al. [8] pointed out that the threshold stress level of GFRP bars in concrete is 25–30% of the ultimate tensile stress. That is the maximum stress at which microcracks of the matrix do not occur. Studies by Almusalam and Al Salloum. [17] and Davaloset al. [18] also reached similar conclusions. Therefore, it is crucial to study the threshold stress levels in the corrosive environment for different kinds of GFRP bars, in order to prolong their service life.

Although the long-term performance of unstressed GFRP bars exposed to corrosion environments has been studied [4,19,20,21], the prediction of stressed GFRP (GFRP bars subjected to a sustained load) bars is still lacking. In general, GFRP bars are always working under stress, and the degradation rate and degradation mechanism of stressed GFRP bars in corrosive environments may be different from those of unstressed GFRP bars [18]. Therefore, it is necessary to carry out a long-term performance study of stressed GFRP bars in corrosive environments.

In summary, the main work of this research is summarized as follows:The residual tensile properties of stressed GFRP bars in a corrosive environment were evaluated and the obtained results were compared with the literature and the design limits given by ACI 440.1R-15 [22];The experimental phenomena and SEM analysis results were analysed to investigate the degradation mechanism of GFRP bars exposed to simulated seawater solution and concrete pore solution;The long-term performance of GFRP bars exposed to concrete pore solution under different stress levels was predicted based on Arrhenius theory.

## 2. Experimental Program

### 2.1. The GFRP Bars Used in this Study Were Prepared

The GFRP bars used in this study were prepared by the pultrusion process with E44 epoxy resin and fibers. The detailed production parameters are shown in Table 1. A 0.4 mm diameter Kevlar fiber (Harbin FRP Institute, Harbin, Heilongjiang, China) was wrapped on the surface of the GFRP bar, which resulted in varied diameters of the GFRP bars at different locations. The nominal diameter of the GFRP bar was 9.2 mm, as shown in Figure 1. The tensile and physical properties of the GFRP bars are listed in Table 2. The tensile properties of the bar were determined by ACI 440.1R-15 [22] and the physical properties were provided by the manufacturer.

### 2.2. Test Parameters

This study mainly includes three parameters:Sustained tensile stresses: 20% and 40% of the ultimate tensile strength were selected in this test. The corresponding strains were 5200 and 10,400 με, respectively. These levels of strain are about 1.68–3.35 times the values recommended by ACI 440.1R-15 for creep rupture strain (Table 2). This was done to explore the material’s potential and evaluate how conservative the current codes and guidelines are;Surrounding environment: Two different corrosive mediums were used to simulate the seawater and concrete environment. The compositions of the two simulated solutions are shown in Table 3, and the pH of the two solutions meets the requirements of ACI 440.3R-04 [23];Temperature: Both ambient and elevated temperatures were used, with the ambient temperature being 23 °C and elevated temperatures being 40 and 60 °C.

### 2.3. Specimen Design and Test Procedure

Test specimen: The total length of the GFRP bar used for testing was 1000 mm, and each end was anchored with 275 mm-long and 18 mm-inner diameter steel tubes for applying axial tension. The middle of the specimens was instrumented with two strain gauges to monitor longitudinal strains. In order to expose the GFRP to a specific solution, a plastic pipe (Linyi Dongli Plastic building materials CO., LTD, Linyi, Shandong, China) with a diameter of 63 mm and a length of 250 mm was inserted into the middle of the bar, which served as a reservoir. A PVC pipe was inserted into the gap between the two ends of the plastic pipe and the GFRP bar, and then glued them together. An opening with a diameter of 15 mm was left in the PVC pipe at one end of the plastic pipe, to allow for frequent replacement of the solution when the pH of the solution was lower than the set value. The unstressed specimen is shown in Figure 2a.

Loading device: The loading device was composed of a reaction force frame and a spring. The reaction frame was connected by two 18 mm thick steel plates and four steel rods (diameter = 8 mm) through bolts, and the stiffness of the spring was 790 kN/mm. The loading procedures mainly consisted of two steps: the first step was to fix the specimen on the reaction frame, and the second step was to compress the spring by constantly tightening the nut (2) to provide a reaction force. Then, the force was applied to the test specimen through the reaction frame until the compression of the spring reached the target length and the strain gauge reached the corresponding strain. Finally, the nut (3) was tightened so that the GFRP bar was under sustained stress. During the test, when the compression length of the spring dropped below the calculated length, it was compressed to the target length. In the subsequent test process, the strains of the GFRP bar were checked every 5 days to observe the creep process. The stressed specimen is shown in Figure 2b.

Temperature control: A glass fiber heating belt (Beijun Group, Yanchen, Jiangsu, China) was used to wrap the plastic tube and heat the solution in the plastic tube. A temperature sensor between the plastic tube and the heating belt was installed, with a US-controlled TN-99 temperature controller (Tianjin, China), to monitor the temperature. During the test, the temperature error was controlled within ±2 °C, as shown in Figure 3.

Test procedure: After reaching a defined duration under the controlled testing condition, the preload on specimens was released and the PVC container was removed. Then, the specimen was subjected to the tensile test according to ACI 440.3R B.2-04 [23].

### 2.4. Specimen Numbering

In this research, three specimens were tested for each condition, and a total of 162 specimens were tested. The specimen numbering was defined as follows.

The numbering of conditioned specimens consists of five parts: The first letter G stands for GFRP, and the second letter stands for the solution type. For example, S stands for seawater solutions, and C stands for concrete pore solutions. Then, the first number represents the exposure temperature, the second number represents the sustained stress level, and the third number represents the exposure time. For a typical conditioned specimen, G/S/40/20/30 indicates that a GFRP bar with a stress level of 20% is exposed to a simulated seawater solution at 40 °C for 30 days. 

### 2.5. Scanning Electron Microscope (SEM)

A JEOL 7001F field emission scanning electron microscope (SEM, Osaka, Japan) was used to observe the cross section of the GFRP bar samples to understand the degradation mechanism of GFRP bars. The samples observed with the SEM were reference specimens and specimens conditioned in 60 °C solutions for the longest duration, which was 90 days. To prepare the samples, the GFRP bars were embedded in epoxy resin, cut with a low-speed saw, and polished with sandpaper and fluffy cloth.

## 3. Results and Discussion

Table 4 and Table 5 list the results obtained for the GFRP bars tested in the laboratory. The results include the residual tensile strength vs. strain, tensile strength vs. strain retention, and the residual elastic modulus. 

### 3.1. Residual Tensile Strength and Strain 

Columns 3 and 4 in Table 4 and Table 5 present the experimental residual tensile strength and corresponding retention for the tested specimens. It can be observed that the tensile strength degradation behavior of GFRP bars exposed to seawater and concrete pore solutions is almost similar. That is, the higher the temperature and stress level, the lower the tensile strength retention. 

The tensile strength retentions of GFRP bars are summarized in Figure 4. It can be concluded from Figure 4 that a higher temperature accelerates the degradation rate of GFRP bars. When the temperature increased from 0 to 60 °C, the tensile strength retention of unstressed GFRP bars decreased by 16.4% and 25.0%, respectively, after being exposed to seawater and concrete pore water solution for 60 days. This indicates that the diffusion rate of the corrosive mediums is accelerated by higher temperatures. The curves of the GFRP bars under stress levels of 0% and 20% were close to each other, as shown in Figure 4. However, when the stress level increased to 40% of the ultimate tensile strength, the degradation rate of GFRP bars was significantly increased, as shown in Figure 4. This indicates that there were no micro-cracks in the matrix at a 20% stress level, and the corrosive mediums passing through the matrix were dominated by diffusion. However, a 40% stress level represents the “medium stress level” defined by Benmokrane et al. [10] This stress level is able to cause micro-cracks in the matrix, and the corrosive mediums can travel through the matrix faster and easier, accelerating the degradation of GFRP bars. The synergistic effect of stress and temperature causes the tensile strength degradation of GFRP bars to become more obvious when exposed to concrete pore water solution. After being exposed to concrete pore solution at 60 °C for 90 days, the tensile strength retention of GFRP bars tested under a 40% stress level is 24.5% lower than those of unstressed GFRP bars, but only 9.4% lower than those exposed to seawater solution. Columns 7 and 8 in Table 4 and Table 5 present the experimental residual tensile strain and corresponding retention for the specimens. It can be observed that the degradation characteristics of tensile strain are similar to those of tensile strength.

### 3.2. Comparing Obtained Results with Literature

In order to verify the applicability of the GFRP bars investigated in this study, this paper compares the test data with results presented in literature. The selected literature includes similar test methods, compositions of solutions, and pH levels, to ensure reliability. Figure 5 compares the tensile strength retention of GFRP bars exposed to seawater solutions (G1 and G2 represent different types of GFRP bars in the literature). From Figure 5, it can be seen that except for the improved GFRP bars (G2) used by Kim et al. [24], the GFRP bars investigated in this study had a better residual tensile strength compared to other GFRP bars reported in the literature, indicating their good resistance to seawater. Figure 6 compares the tensile strength retention of GFRP bars exposed to concrete pore water solution. It can be seen from Figure 6 that, with or without stress, the tensile strength retention of GFRP bars used in this study is higher than that of the GFRP bars used in the literature for the same test duration. The degradation rate of GFRP bars used by Sen et al. [20] is very fast, and the degradation rate has already been increased significantly under a stress level of 15%, as shown in Figure 6. The reason for this situation is that Sen et al. [20] used GFRP bars with smaller diameters, so the corrosion mediums could penetrate into the GFRP bars more rapidly. Another reason is that the GFRP bars used were an early FRP product with low manufacturing maturity.

### 3.3. Residual Modulus of Elasticity

Due to the low elastic modulus of GFRP bars, under the same reinforcement ratio, GFRP-RC members tend to exhibit larger deformation and result in wider cracks compared to steel-RC members [25,26,27]. The design of GFRP-RC members is usually governed by the serviceability limit state (deformation and cracking). Therefore, the reduction in the elastic modulus of GFRP bars during the service life is also an important factor. Figure 7 compares the elastic modulus of GFRP bars exposed to seawater and concrete pore water solutions at 23, 40, and 60 °C for 90 days at different stress levels. As shown in Figure 7, the elastic modulus of GFRP bars exposed to seawater solution and concrete pore aqueous solution is 0.95–1.04 times and 0.94–1.03 times that of unstressed GFRP bars, respectively. This result indicates that the sustained loads (20% and 40% stress levels) had almost no effect on the elastic modulus of GFRP bars, and degradation of the tensile strength and strain of GFRP bars remained at the same level, which agrees with previous literature [9,16,28].

### 3.4. Comparison with ACI 440.1R-15

Creep tensile strain: According to the changes in the strain of GFRP bars over time at sustained stress, the creep characteristics can be analysed. Figure 8 shows this change for GFRP bars exposed to seawater and concrete pore solution at 60 °C. It can be observed that almost no significant changes in the strain occurred after 90 days. At the 20% and 40% stress level, the maximum increase in strain was only 3% and 7% of the initial strain, respectively. Nkurunziza et al. [16] and Robert et al. [6] forecasted the creep strain over the service life of concrete structures (75 years). It was found from their test results that the strain of GFRP bars only experienced an 8% increase in strain at the 38% stress level. This shows that the ACI 440.1R-15 [22] is very conservative in setting the creep failure stress of GFRP bars to 0.2 *f*_fu_ (*f*_fu_ is the design tensile strength of GFRP).

Tensile strength and strain: Figure 9 shows the ratios between the residual tensile strength/strain of conditioned GFRP bars and the design limits specified by ACI 440.1R-15 [22]. It can be seen from Figure 9 that, except for the GFRP bars at the 40% stress level of 90-day exposure to concrete pore solution, the residual tensile strength and tensile strain are slightly lower than the design limits, and the other conditioned GFRP bars have higher values that exceed the design limits. In fact, in the design of a GFRP-RC structure, the GFRP bars will remain below a stress level of 40%, which also proves the applicability of the GFRP bars used in this test.

### 3.5. Microstructure Analysis

The degradation rate of GFRP bars is relatively more rapid when exposed to a solution at 60 ° C. Therefore, only the microstructure analysis of GFRP bars after 90 days of exposure to a solution at 60 ° C was performed to understand the degradation mechanism of GFRP bars. Figure 10 shows the cross-sections of unconditioned GFRP bars at three magnifications (500×, 1000×, and 2000×). It can be seen from Figure 10 that there are no visible voids (debonding) between the fibers and the resin. For the conditioned GFRP bars, as shown in Figure 11, there is a distinct increase in the number of voids between the fibers and the resin, and the width and distribution range of the voids increase with the stress level. Compared with Figure 10, it can be found that Figure 11 shows that the glass fibers of the conditioned GFRP bars have not been significantly damaged, which indicates that the degradation of the GFRP bars in this study mainly occurred at the fiber/matrix interface.

## 4. Prediction of the Long-Term Performance of GFRP Bars

### 4.1. Prediction Model Selection

The long-term performance prediction models for commonly used FRP composites are based on Arrheneius acceleration theory [4,18], and several prediction models have been proposed in the literature. After a careful analysis and comparison of the existing models [18,20,21], the prediction model proposed by Phani and Bose [29] could most accurately reflect the degradation process of FRP bars. This model was first used to predict the degradation of flexural strength of inter-laminar plates in wet and high-temperature environments. Moreover, the degradation mechanism for the model is assumed to be debonding at the fiber/matrix interface. This has also been confirmed by the SEM test results presented in this paper. The model is shown in Equation (1):(1)$$Y=(100-Y_{\infty })\exp(-t/\tau )+Y_{\infty }$$
where *Y* is the tensile strength retention (%), *t* is the exposure time, *τ* is the fitted parameter, and *Y*_∞_ is the tensile strength retention (%) at the exposure time of infinity. As the GFRP bars were exposed to the solution for only a short time in this study, the tensile strength of the GFRP bars was still decreasing after exposure, so it is impossible to know the tensile strength retention *Y*_∞_ after an infinite exposure time. Therefore, Equation (2) was used to predict the long-term performance of GFRP bars. Previous research (Chen et al. [4] and Wu et al. [19]) has confirmed the applicability and reliability of Equation (2) for unstressed FRP bars.
(2)Y=100exp(−t/τ)

### 4.2. Arrheneius Theory

According to Arrheneius theory, the relationship between the degradation rate and the temperature can be expressed as
(3)$$k=A\exp(\frac{-E_{a}}{RT})$$
where *k* is the degradation rate (1/time), *A* is a constant related to the material degradation process, *E*_a_ is the activation energy, *R* is the universal gas constant, and *T* is the temperature in Kelvin.

It should be noted that the primary assumption of Arrheneius theory is that the single dominant degradation mechanism of the material does not change with time and temperature during the exposure, but the rate of degradation is accelerated with temperature [30]. Therefore, Equation (3) can be rewritten as
(4)$$\ln(\frac{1}{k})=\frac{E_{a}}{R}\frac{1}{T}-\ln(A)$$

From Equation (4), the logarithm of time needed for a material property to reach a given value is a linear function of 1/*T* with the slope of the *E*_a_/*R* value.

According to Equation (3), the acceleration coefficient (*AF*) for a range of high temperature (*T*_1_) to low temperature (*T*_0_) can be calculated as
(5)$$AF=\frac{t_{0}}{t_{1}}=\frac{c/k_{0}}{c/k_{1}}=\frac{k_{1}}{k_{0}}=\frac{A\exp(-E_{a}/RT_{1})}{A\exp(-E_{a}/RT_{0})}exp\left[\frac{E_{a}}{R}(\frac{1}{T_{0}}-\frac{1}{T_{1}})\right]$$
where *AF* is the acceleration coefficient; *t*_1_ and *t*_0_ are the times required to reach a given tensile strength retention value at temperatures of *T*_1_ and *T*_0_, respectively; *c* is a constant; and *k*_1_ and *k*_0_ are the degradation rates at temperatures of *T*_1_ and *T*_0_, respectively.

Therefore, the long-term tensile performance of GFRP bars at any given temperature can be predicted using Equation (2).

### 4.3. Prediction Procedure

The long-term performance prediction of GFRP bars consists of four steps. Step 1: As shown in Figure 12, the obtained experimental data (Figure 4b) could be fit using Equation (2) and the linear regression coefficient *τ* and correlation coefficient *R*^2^ were as shown in Table 6.

Step 2: The parameter *τ* (Table 6) could be substituted in Equation (2) to obtain the time required to achieve strength retention of 50%, 60%, 70%, and 80% at 23, 40, and 60 °C under different stress levels. Then, the Arrhenius relationships were obtained by plotting the natural logarithm of time (*t*) against the inverse of temperature (1/*T*), and the data were fit using Equation (4), as shown in Figure 13. The fitted *E*a/*R* and correlation coefficient *R*^2^ are presented in Table 7. It was found that the correlation coefficients of all regression lines were close to 1, and the straight lines at different temperatures were parallel to each other. This indicates that the Arrhenius relationship can be used to describe the degradation rate of GFRP bars, as the degradation mechanism may not change with temperature and time during exposure in the range test.

Step 3: 20 °C was taken as a reference temperature and Equation (5) was used to calculate the *AF* values for other temperatures (23, 40, and 60 °C) based on the reference temperature of 20 °C. The obtained *AF* values are listed in Table 8.

Step 4: Once the *AF*s for 23, 40, and 60 °C were obtained, Figure 4b was transformed into Figure 14 by multiplying the exposure time at 23, 40, and 60 °C with the corresponding *AF*s. Then, Equation (2) was used to fit Figure 9 to obtain the master curves for tensile strength retention versus exposure time at 20 °C under different stress levels. The fitting parameters and correlation coefficients are listed in Table 9. It is worth noting that the *τ* values in Table 9 are very close to the *τ* values obtained at 23 °C presented in Table 6. This close correlation also confirms the validity of this procedure for predicting the long-term performance of GFRP bars. Therefore, the tensile strength retention of the unstressed and stressed GFRP bars at any exposure time could be predicted using Equation (4) and the *τ* values listed in Table 9.

### 4.4. Discussion on Predicted Results

As shown in Figure 14, it can be can that the time required to achieve 50% tensile strength retention under three stresses (0%, 20%, and 40%) at 20 °C in concrete pore solution is 1.5, 1.2, and 0.65 years, respectively. Mufti et al. [31] found that there is no degradation for GFRP bars in a GFRP-RC bridge with 5–8 years of service by using analytical methods (e.g., SEM). It can be seen that the long-term performance results of GFRP bars directly exposed to corrosive solution can not truly reflect the degradation of GFRP bars in field concrete structures. Therefore, long-term data should be collected in field applications, and the correlation between the degradation of GFRP bars in accelerated tests and field applications needs to be investigated.

From the test results in Table 5 and Figure 4, it can be seen that a stress level of 20% does not seem to affect the degradation rate of GFRP bars. However, the degradation rates significantly increase with time, as shown in Figure 11. For strength retention of 50%, the required time under a 20% stress level is 25% shorter than that of unstressed GFRP bars. Davaloset al. [18] also found a similar phenomenon in their study on the durability of GFRP bars embedded in concrete. This may be due to the short exposure time of GFRP bars.

The activation energy (*E*_a_) is defined as the energy required for a chemical reaction. The lower the required activation energy of a GFRP bar, the faster the GFRP bar degradation rate. For example, the activation energy of GFRP bars at a 40% stress level is 28.6% and 14% lower compared to lower stress levels of 0% and 20%, respectively, as shown in Table 7. The required activation energy of GFRP bars exposed to alkaline solution ranges from 12 to 41 KJ/mol [4,20,21,28], while the required activation energy of GFRP bars embedded in concrete ranges from 36 to 91 KJ/mol [7,18,32]. Therefore, the degradation rate of GFRP bars in concrete is much lower than those directly exposed to corrosive solution. It also shows that the long-term performance prediction results are conservative based on laboratory-simulated testing.

## 5. Conclusions

This experimental research has investigated the durability of new GFRP bars exposed to laboratory-simulated seawater and concrete environments. The main conclusions are as follows:(1)The degradation rate of GFRP bars in seawater solution is significantly lower than that in concrete pore solution, and the difference in degradation rate becomes more obvious as the temperature and stress level increase;(2)A stress level of 40% is able to cause micro-cracks in the resin matrix of the GFRP bars, accelerating the degradation rate of the GFRP bar. In the design of GFRP-RC structures, the stress level in the GFRP bars should be limited to under 40% of the ultimate tensile strength;(3)It has been found that a lower stress level of 20% does not affect the degradation rate of GFRP bars. However, based on the prediction analysis of the service life of GFRP bars, it has been found that for the same tensile strength retention, the time required for a stress level of 20% is significantly reduced compared to a 0% stress level. With the increase of the exposure time of GFRP bars, the damage of GFRP bars increases continuously and the degradation rate of damage GFRP bars is accelerated by a 20% stress level;(4)The elastic modulus of GFRP bars is not affected by the corrosive environment and stress level, and the degradation rate for tensile strength and tensile strain tends to remain at the same level;(5)The residual tensile properties of all GFRP bars can meet the requirements of ACI 440.1R-15, except for those tested under a 40% stress level in 60 °C concrete pore solution for 90 days;(6)Since the GFRP bars were directly exposed to the corrosion solution, the long-term performance prediction results of the GFRP bars obtained using Arrheneius theory are considered to be conservative. Future research is needed to establish the correlation between the degradation of GFRP bars in laboratory-simulated and field conditions, in order to provide practical guidelines for GFRP-RC structures.

## Figures and Tables

**Figure 1 materials-13-02341-f001:**
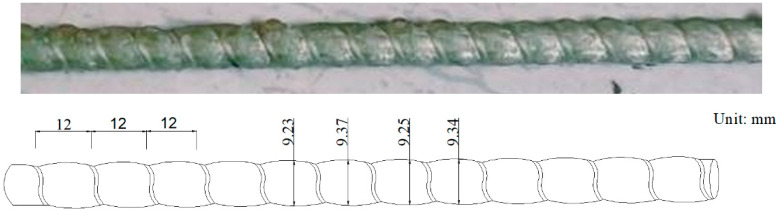
GFRP bars.

**Figure 2 materials-13-02341-f002:**
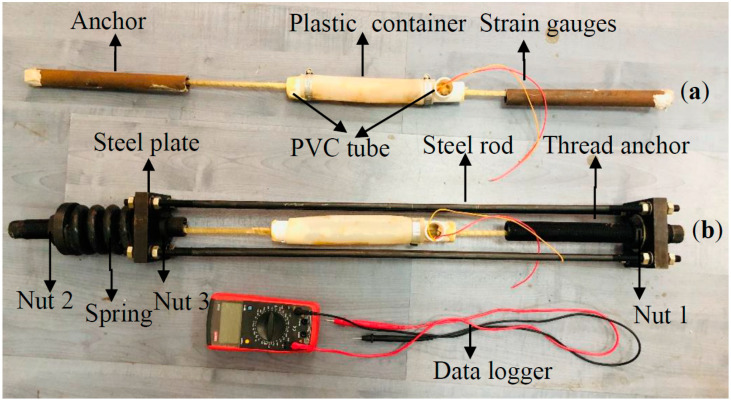
Test specimens: (**a**) Unstressed specimens and (**b**) stressed specimens.

**Figure 3 materials-13-02341-f003:**
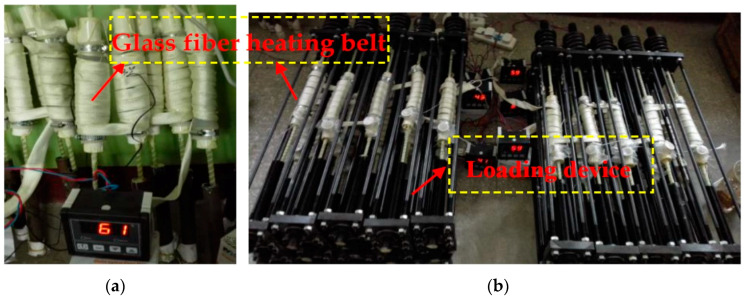
Temperature control measures: (**a**) Unstressed specimens and (**b**) stressed specimens.

**Figure 4 materials-13-02341-f004:**
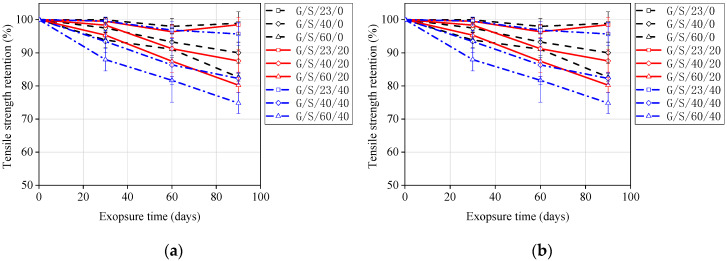
Tensile strength retention of GFRP bars exposed to (**a**) seawater solution and (**b**) concrete pore solution.

**Figure 5 materials-13-02341-f005:**
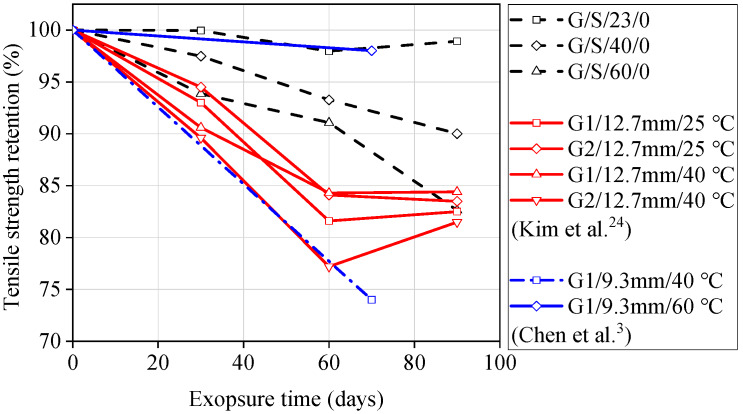
Comparison of experimental data (GFRP bars exposed to seawater solution) from this study and previous literature.

**Figure 6 materials-13-02341-f006:**
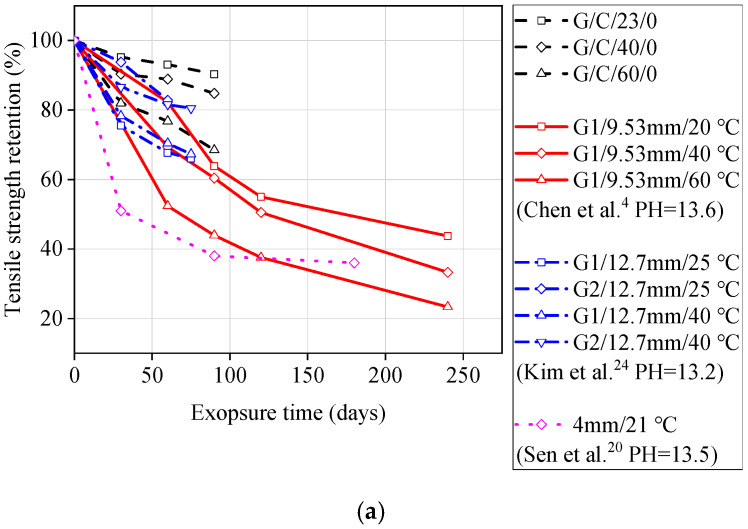
Comparison of experimental data (GFRP bars exposed to concrete pore solutions solution) from this study and previous literature: (**a**) unstressed bars; (**b**) stressed bars.

**Figure 7 materials-13-02341-f007:**
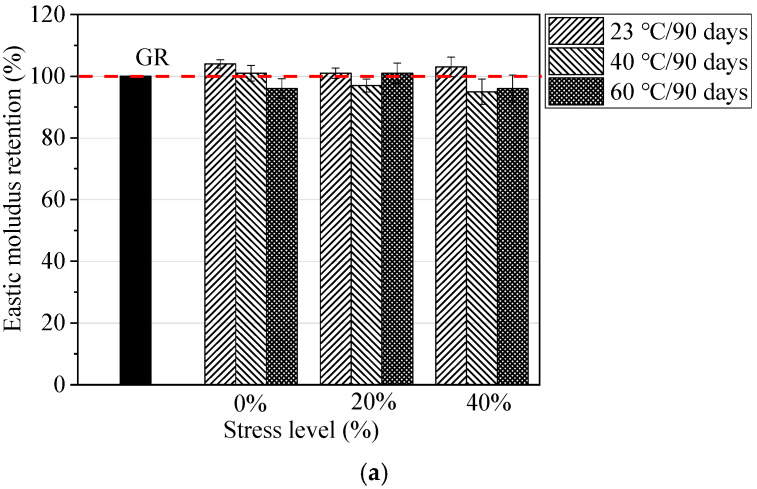
Elastic modulus retention of conditioned GFRP bars: (**a**) Exposure to seawater solutions and (**b**) exposure to concrete pore solutions.

**Figure 8 materials-13-02341-f008:**
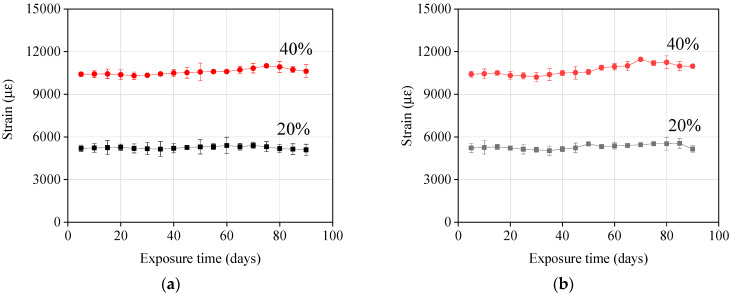
Change in axial strain in GFRP bars over time under different stress levels: (**a**) Exposure to seawater solutions and (**b**) exposure to concrete pore solutions.

**Figure 9 materials-13-02341-f009:**
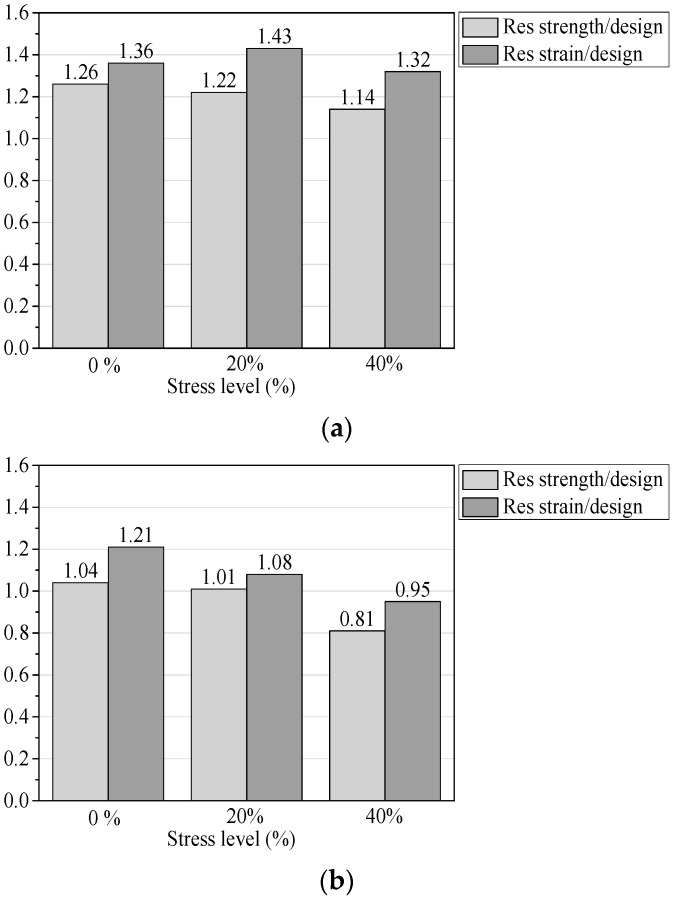
Ratios between the residual tensile strength/strain of conditioned GFRP bars and design limits specified by ACI 440.1R-15 (**a**) Exposure to seawater solutions and (**b**) exposure to concrete pore solutions.

**Figure 10 materials-13-02341-f010:**
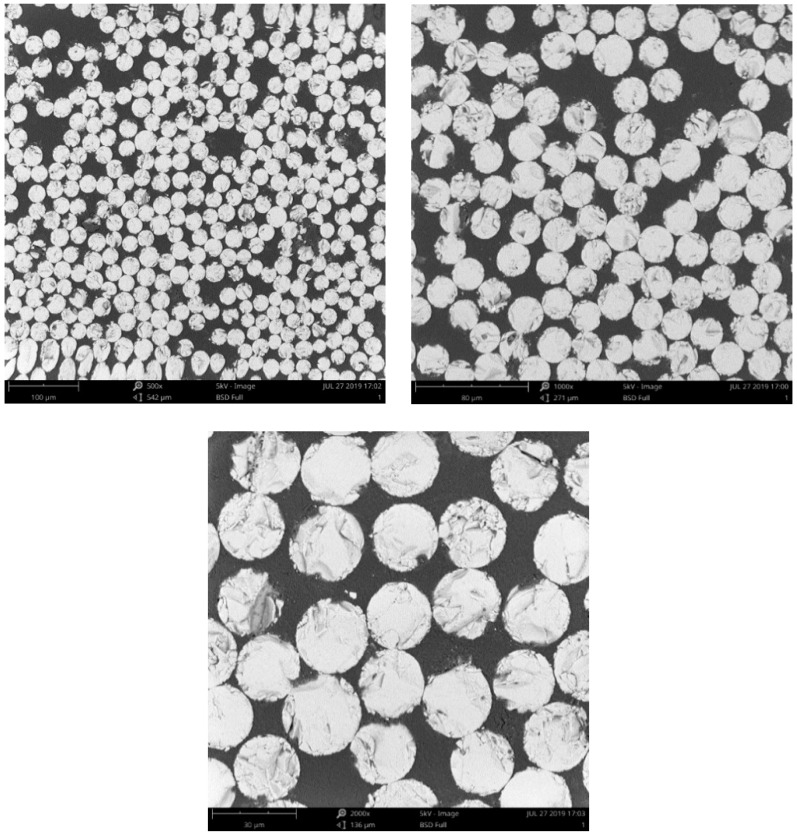
SEM images of the cross sections of the unconditioned GFRP bars.

**Figure 11 materials-13-02341-f011:**
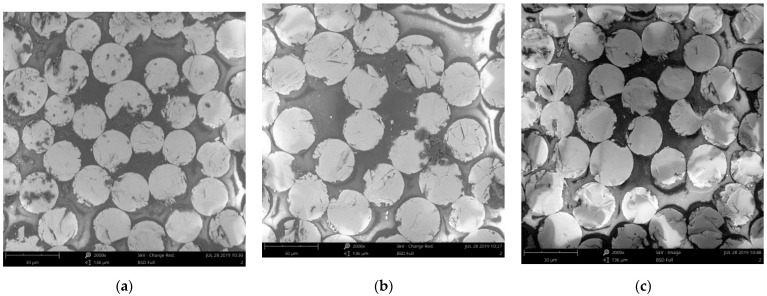
SEM images of the cross sections of the conditioned GFRP bars:(**a**) G/S/23/0/90, (**b**) G/S/23/20/90; (**c**) G/S/23/40/90; (**d**) G/C/23/0/90; (**e**) G/C/23/20/90; (**f**) G/C/23/40/90.

**Figure 12 materials-13-02341-f012:**
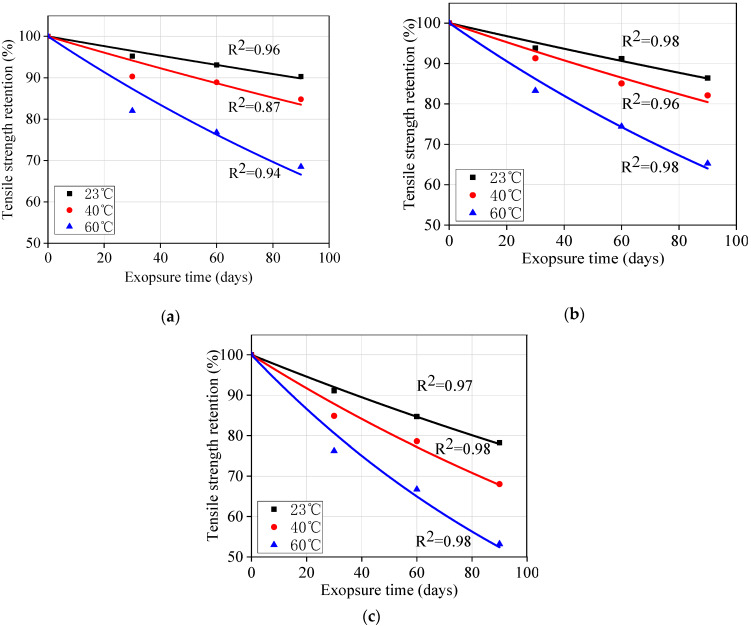
Fitted curves for tensile strength retention versus time: (**a**) 0% stress level; (**b**) 20% stress; (**c**) 40% stress level.

**Figure 13 materials-13-02341-f013:**
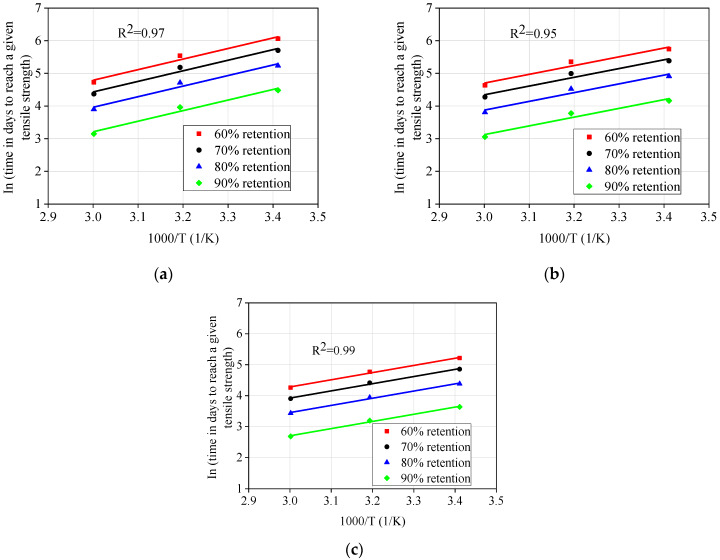
Arrhenius plots of tensile strength degradation: (**a**) 0% stress level, (**b**) 20% stress level, and (**c**) 40% stress level.

**Figure 14 materials-13-02341-f014:**
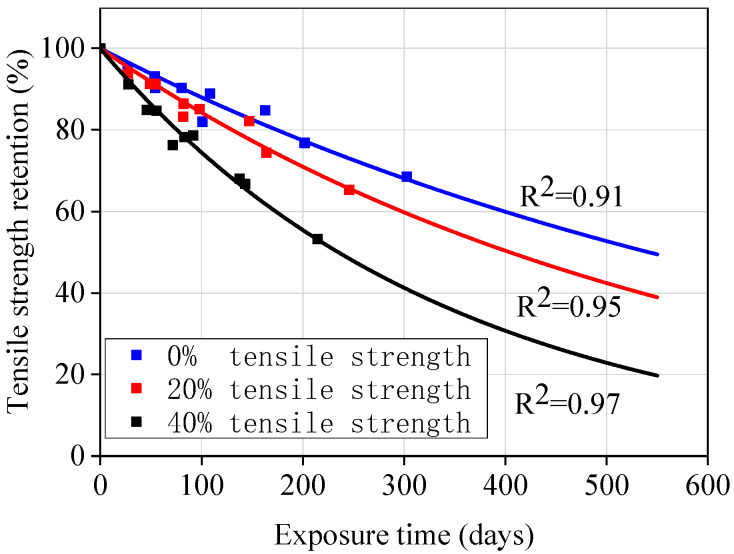
Master curves for GFRP bars exposed to concrete pore solutions at 20 °C.

**Table 1 materials-13-02341-t001:** Production parameters of fiber-reinforced polymer (FRP) bars.

Type	Nominal Diameter (mm)	Reinforced Fiber					Resin Matrix	
		Brand	Diameter (μm)	Tensile Strength (MPa)	Tensile Modulus (GPa)	Density (g/cm^3^)	Epoxy	Hardener
GFRP	8	E-glass 9600 Tex	28	2250	82	2.66	Bisphenol-A E44	MeHHPA ^a^

^a^ MeHHPA represents Methylhexahydrophthalic Anhydride.

**Table 2 materials-13-02341-t002:** Tensile and physical properties of reference glass-fiber-reinforced polymer (GFRP) bars.

	Property	Symbol	GFRP Bars
Tensile properties	Ultimate tensile strength (MPa)	*f* _u,ave_	1200 ± 25
	Guaranteed tensile strength (MPa)	*f**_fu_ = *f*_u,ave_−3σ	1125
	Environmental reduction factor (ACI 440.1R-15)	*C* _E_	0.7
	Design tensile strength (ACI 440.1R-15) (MPa)	*f*_fu_ = *C*_E_ *f**_fu_	787.5
	Modulus of elasticity (GPa)	*E* _f,ave_	45 ± 0.8
	Ultimate strain (%)	*ε* _u,ave_	2.6 ± 0.12
	Guaranteed strain (%)	*ε**_u_ = *ε*_u, ave_−3σ	2.24
	Design strain (%)	*ε*_u_ = *C*_E_*ε**_u_	1.57
	Creep strain (%)	20%*ε*_u_	0.31
Physical properties	Glass transition temperature (°C)	*T* _g_	140
	Fiber content by volume (%)	*V* _f_	71.2
	Fiber content by weight (%)	*W* _f_	82.5
	Transverse coefficient of thermal expansion (×10^−6^/°C)	*α* _T_	23.2

**Table 3 materials-13-02341-t003:** Compositions of simulated seawater and concrete pore solutions.

Solution Type	Quantities (Gram Per Liter)					pH
seawater	NaCl	MgCl_2_	Na_2_SO_4_	CaCl_2_	KCl	8.1
	24.53	5.20	4.10	1.16	0.71	
concrete pore solutions	NaOH	KOH	Ca(OH)_2_	-	-	13.4
	2.11	19.63	2.10	-	-	

**Table 4 materials-13-02341-t004:** Experimental tensile properties of conditioned GFRP specimens exposed to seawater solution.

Specimen	Applied Stress, % *f*_u,ave_	Specimen (Tension Test)				Strain (Tension Test)				Residual Elastic Modulus (GPa)
		Residual (MPa)	*f*_res_/*f*_u,ave_ (%)	*f*_res_/*f**_fu_ (%)	*f*_res_/*f*_fu_ (%)	Residual *με*_res_ (%)	*ε_res_/ε*_u,ave_ (%)	*ε_res_/ε**_u_ (%)	*ε_res_/ε*_u_ (%)	
G/S/23/0/30	0	1199.4	99.95	106.61	1.52	2.64	99.54	117.86	168.15	45.45
G/S/23/0/60		1175.64	97.97	104.50	1.49	2.69	98.46	120.09	171.34	43.65
G/S/23/0/90		1186.92	98.91	105.50	1.51	2.54	97.69	113.39	161.78	46.8
G/S/40/0/30		1169.76	97.48	103.97	1.49	2.65	101.92	118.3	168.79	44.1
G/S/40/0/60		1119.24	93.27	99.48	1.42	2.51	96.54	112.05	159.87	44.55
G/S/40/0/90		1080.12	90.01	96.01	1.37	2.38	91.54	106.25	151.59	45.45
G/S/60/0/30		1126.32	93.86	100.1	1.43	2.58	99.23	115.18	164.33	43.65
G/S/60/0/60		1092.84	91.07	97.14	1.39	2.53	97.31	112.95	161.15	43.2
G/S/60/0/90		991.56	82.63	88.13	1.26	2.14	82.31	95.54	136.31	46.35
G/S/23/20/30	20	1194.24	99.52	106.15	1.52	2.63	101.15	117.41	167.52	45.45
G/S/23/20/60		1156.08	96.34	102.76	1.47	2.49	95.77	111.16	158.6	46.35
G/S/23/20/90		1181.64	98.47	105.03	1.5	2.6	100	116.07	165.61	45.45
G/S/40/20/30		1180.32	98.36	104.91	1.5	2.76	106.15	123.21	175.8	42.75
G/S/40/20/60		1094.76	91.23	97.31	1.39	2.39	91.92	106.7	152.23	45.9
G/S/40/20/90		1050.72	87.56	93.39	1.33	2.41	92.69	107.59	153.5	43.65
G/S/60/20/30		1143.84	95.32	101.67	1.45	2.59	99.62	115.63	164.97	44.1
G/S/60/20/60		1049.4	87.45	93.28	1.33	2.29	88.08	102.23	145.86	45.9
G/S/60/20/90		962.76	80.23	85.57	1.22	2.25	86.54	100.45	143.31	42.75
G/S/23/40/30	40	1195.08	99.59	106.22	1.52	2.71	104.23	120.98	172.61	44.1
G/S/23/40/60		1162.2	96.85	103.30	1.48	2.66	102.31	118.75	169.43	43.65
G/S/23/40/90		1148.04	95.67	102.04	1.46	2.66	102.31	118.75	169.43	43.2
G/S/40/40/30		1121.04	93.42	99.64	1.42	2.42	93.08	108.04	154.14	46.35
G/S/40/40/60		1035.84	86.32	92.07	1.32	2.33	89.62	104.02	148.41	44.55
G/S/40/40/90		987.24	82.27	87.75	1.25	2.17	83.46	96.88	138.22	45.45
G/S/60/40/30		1055.4	87.95	93.81	1.34	2.5	96.15	111.61	159.24	42.3
G/S/60/40/60		980.16	81.68	87.12	1.24	2.29	88.08	102.23	145.86	42.75
G/S/60/40/90		897.72	74.81	79.79	1.14	2.08	80	92.86	132.48	43.2

**Table 5 materials-13-02341-t005:** Experimental tensile properties of conditioned GFRP specimens exposed to concrete pore solution.

Specimen	Applied Stress, % *f*_u,ave_	Specimen (Tension Test)				Strain (Tension Test)				Residual Elastic Modulus (GPa)
		Residual (MPa)	*f*_res_/*f*_u,ave_ (%)	*f*_res_/*f**_fu_ (%)	*f*_res_/*f*_fu_ (%)	Residual *με*_res_ (%)	*ε_res_/ε*_u,ave_ (%)	*ε_res_/ε**_u_ (%)	*ε_res_/ε*_u_ (%)	
G/C/23/0/30	0	1142.4	95.2	101.55	145.07	94.36	96.54	112.05	159.87	45.45
G/C/23/0/60		1117.2	93.1	99.31	141.87	95.11	97.31	112.95	161.15	44.1
G/C/23/0/90		1083.6	90.3	96.32	137.6	91.35	93.46	108.48	154.78	44.55
G/C/40/0/30		1083.6	90.3	96.32	137.6	93.23	95.38	110.71	157.96	43.65
G/C/40/0/60		1066.8	88.4	94.83	135.47	87.22	89.23	103.57	147.77	45.9
G/C/40/0/90		1017.6	85.8	90.45	129.22	82.71	84.62	98.21	140.13	46.35
G/C/60/0/30		984	82	87.47	124.95	79.7	81.54	94.64	135.03	46.35
G/C/60/0/60		921.6	75.8	81.92	117.03	78.57	80.38	93.3	133.12	44.1
G/C/60/0/90		822	67.8	73.07	104.38	71.43	73.08	84.82	121.02	43.2
G/C/23/20/30	20	1125.6	93.8	100.05	142.93	90.6	92.69	107.59	153.5	46.8
G/C/23/20/60		1094.4	91.2	97.28	138.97	96.24	98.46	114.29	163.06	42.75
G/C/23/20/90		1036.8	86.4	92.16	131.66	85.71	87.69	101.79	145.22	45.45
G/C/40/20/30		1095.6	91.3	97.39	139.12	88.72	90.77	105.36	150.32	46.35
G/C/40/20/60		1020.84	85.07	90.74	129.63	86.09	88.08	102.23	145.86	44.55
G/C/40/20/90		985.32	82.11	87.58	125.12	84.96	86.92	100.89	143.95	43.65
G/C/60/20/30		998.88	83.24	88.79	126.84	86.84	88.85	103.13	147.13	43.2
G/C/60/20/60		892.92	74.41	79.37	113.39	75.94	77.69	90.18	128.66	44.1
G/C/60/20/90		782.76	65.23	69.58	1.01	64.29	65.77	76.34	108.92	45.9
G/C/23/40/30	40	1093.56	91.13	97.21	138.86	87.97	90	104.46	149.04	46.8
G/C/23/40/60		1016.64	84.72	90.37	129.1	84.21	86.15	100	142.68	45.45
G/C/23/40/90		938.88	78.24	83.46	119.22	77.82	79.62	92.41	131.85	45.45
G/C/40/40/30		1018.56	84.88	90.54	129.34	87.59	89.62	104.02	148.41	43.65
G/C/40/40/60		943.44	78.62	83.86	119.8	81.95	83.85	97.32	138.85	43.2
G/C/40/40/90		816.48	68.04	72.58	103.68	72.56	74.23	86.16	122.93	42.3
G/C/60/40/30		914.64	76.22	81.3	116.14	75.56	77.31	89.73	128.03	45.45
G/C/60/40/60		800.76	66.73	71.18	101.68	68.42	70	81.25	115.92	44.1
G/C/60/40/90		638.52	53.21	56.76	81.08	56.02	57.31	66.52	94.9	42.75

**Table 6 materials-13-02341-t006:** Coefficients of regression in Equation (2).

Temperature (°C)	0 Stress Level		20% Stress Level		40% Stress Level	
	*τ*	*R* ^2^	*τ*	*R* ^2^	*τ*	*R* ^2^
23	840	0.96	610	0.98	361	0.99
40	499	0.87	413	0.99	232	0.98
60	221	0.94	202	0.99	139	0.98

**Table 7 materials-13-02341-t007:** Coefficients of regression equations for Arrhenius plots.

Tensile Strength Retention (%)	0% Stress Level			20% Stress Level			40% Stress Level		
	*E*_a_/*R*	*R* ^2^	*E*_a_ (KJ/mol)	*E*_a_/*R*	*R* ^2^	*E*_a_ (KJ/mol)	*E*_a_/*R*	*R* ^2^	*E*_a_ (KJ/mol)
50	3235	0.97	26.9	2680	0.99	22.3	2317	0.99	19.2
60	3235	0.97	26.9	2680	0.99	22.3	2317	0.99	19.2
70	3235	0.97	26.9	2680	0.99	22.3	2317	0.99	19.2
80	3237	0.97	26.9	2680	0.99	22.3	2317	0.99	19.2

**Table 8 materials-13-02341-t008:** Values for acceleration factors.

Temperature (°C)	0% Stress Level	20% Stress Level	40% Stress Level
23	0.894	0.912	0.923
40	1.809	1.635	1.529
60	3.364	2.733	2.385

**Table 9 materials-13-02341-t009:** Coefficients of the regression equations for the master curves.

Stress Level (%)	*τ*	*R* ^2^
0	781	0.92
20	583	0.96
40	339	0.98

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
