# Peer review of "Prediction of the Long-Term Performance and Durability of GFRP Bars under the Combined Effect of a Sustained Load and Severe Environments"

_materials, 2020, doi:10.3390/ma13102341_

Round 1

Reviewer 1 Report

Lines 38/39: Please explain abbreviations GFRP, AFRP and CFRP

Line 86: Correct: "The GFRP bars used in this study were prepared.."

Table 2: What are "Guaranteed strain" and "Design strain"?

Line 101: I do not understand corresponding strains. 5200με and 10400μ??

No space here: "60°C".

Reference should be, e.g. "Robert et al."and not "Robert et. al".

Author Response

We greatly appreciate you for reviewing our manuscript. Each comment will be very helpful for us to improve our writing skills. All issues have been answered in the responses and modified in the revised manuscript according to your comments.

Point 1: Lines 38/39: Please explain abbreviations GFRP, AFRP and CFRP

Response 1: GFRP represents glass fiber-reinforced polymer, AFRP represents aramid fiber-reinforced polymer, CFRP represents carbon fiber-reinforced polymer. Theses full names have been added on lines 38-39 of the revised manuscript.

Point 2: Line 86: Correct: "The GFRP bars used in this study were prepared.

Response 2: Has been replaced in the revised manuscript. (line 86)

Point 3: Table 2: What are "Guaranteed strain" and "Design strain"?

Response 3: Guaranteed stain is guaranteed rupture strain of GFRP bar defined as the mean tensile strain at failure of a sample of test specimens minus times the standard deviations (ε*u =εu,ave - 3σ). Design strain is design rupture stain of GFRP bar ,considering reductions for service environment.

Point 4:Line 101: I do not understand corresponding strains. 5200με and 10400μ?

Response 4: 5200 μ ε and 10400μ correspond to the strains of GFRP Bars under % 20 and 40% ultimate tensile strength

Point 5: No space here: "60°C".

Response 5: This error has been corrected in the revised manuscript. (The revisions in the manuscript are marked in red)

Point 6: Reference should be, e.g. "Robert et al."and not "Robert et. al.

Response 6: This error has been corrected in the revised manuscript. (The revisions in the manuscript are marked in red)

Reviewer 2 Report

Overview:

The authors demonstrate an empirical study of durability of glass fiber reinforced polymer bars as a function of time, temperature, tension, and solution conditions. The manuscript is generally well-written with a generally clear preparation and characterization methodology. The following comments are offered to strengthen the quality of the final manuscript.

General Comments:

Line 23: It is recommended that the word “proved” is not used, but another word such as “demonstrated.”

Line 39: Define acronyms AFRP and CFRP

Line 46: In this sentence and throughout the manuscript “PH” is used whereas it should be “pH”

Line 79: For completeness and clarity for a broad audience, recommend defining ACI 440.1R-15.

Table 1: Superscript 3 in Density (g/cm3). Define MeHHPA

Figure 1: The units are identified as mm in the upper right of the figure with bar diameters indicated as ~9mm, however the lateral units of 1.2 in the upper left do not appear to be the same scale.

Line 117: How frequently was the solution replaced?

Figure 2: The caption specifies “(a)” and “(b)” but these are not designated in the figure photo.

Line 143-146: The first number in the sample designation is defined as stress level, however, the number in the example on Line 145 is “23” which corresponds to the ambient temperature condition. Further, the “23” does not match the “20” in Line 146. Please clarify.

Line 148-149: Please identify the SEM manufacturer and model.

Line 168: What “curves” are being referenced?

Table 4 and 5: In the sample designation definition in Lines 143-146, the second number is defined as temperature. Given this designation the first set of samples in each table would have been at 0 deg C which was not described in the Methods.

Section 3.2 G1 and G2 GFRP bars are referenced. How are G1 and G2 different from the bars in this study? This difference or similarity is necessary to place the tensile strength retention comparisons in context.

Results and Discussion. There is no comparison of the GFRP bars to conventional iron bars. Such a comparison might help put the results in context.

Figure 8: (a) and (b) panels are not designated. Are these single sample figures (there are no error bars)?

Section 3.5. How were the cross-section samples prepared? This information should be included in the Methods section.

Equation (4) is described as the “logarithm of time” where it is actually the logarithm of the rate constant.

The logic of the Equation (5) description to the last sentence in Section 4.2 is unclear as to the connection to Equation 2.

Section 4.3 again refers to the “logarithm of time” with respect to Equation 4.

Author Response

We greatly appreciate you for reviewing our manuscript. Each comment will be very helpful for us to improve our writing skills. All issues have been answered in the responses and modified in the revised manuscript according to your comments.

Point 1: It is recommended that the word “proved” is not used, but another word such as “demonstrated.

Response 1: It has been modified in the revised manuscript. (line 23)

Point 2: Define acronyms AFRP and CFRP

Response 2: AFRP represents aramid fiber-reinforced polymer, CFRP represents carbon fiber-reinforced polymer. Theses full names have been added on lines 38-39 of the revised manuscript.

Point 3: Line 46: In this sentence and throughout the manuscript “PH” is used whereas it should be “pH”.

Response 3: This error has been corrected in the revised manuscript. (line 47 and table3)

Point 4: Line 79: For completeness and clarity for a broad audience, recommend defining ACI 440.1R-15.

Response 4: ACI 440.1R-15 is defined as American code for FRP-RC structural design. This definition has been added to the revised manuscript (line79).

Point 5: Table 1: Superscript 3 in Density (g/cm3). Define MeHHPA

Response 5: g/cm3 has been modified to g/cm3. MeHHPA represents Methylhexahydrophthalic Anhydride. This explanation has been added to the revised manuscript (Table.1)

Point 6: Figure 1: The units are identified as mm in the upper right of the figure with bar diameters indicated as ~9mm, however the lateral units of 1.2 in the upper left do not appear to be the same scale.

Response 6: This is a mistake. 1.2 should be changed to 12. This error has been corrected in the revised manuscript (Figure.1 ).

Point 7: Line 117: How frequently was the solution replaced?

Response 7: During the test, the pH value of the solutions was held constant according to regular inspection and solution supplement. These sentences in the revised manuscript has been modified to “An opening with a diameter of 15mm is left in the PVC pipe at one end of the plastic pipe to allow for frequent replacement of the solution when the pH of the solution is lower than the set value” (line 121-122)

Point 8: Figure 2: The caption specifies “(a)” and “(b)” but these are not designated in the figure photo.

Response 9: The specifies “(a)” and “(b)” have been added to the Figure 2.

Point 9: Line 143-146: The first number in the sample designation is defined as stress level, however, the number in the example on Line 145 is “23” which corresponds to the ambient temperature condition. Further, the “23” does not match the “20” in Line 146. Please clarify.

Response 9: This is a mistake in writing. The first number in the sample designation should be defined exposure temperature and the second number defined as the sustained stress level. These relevant errors have been modified in the revised manuscript (line 149-150).

Point 10: Line 148-149: Please identify the SEM manufacturer and model

Response 10: The SEM model is JEOL7001F, made in Japan. This note has been added to the revised manuscript. (line 154)

Point 11: Line 168: What “curves” are being referenced?

Response 11: The sentence in the paper “The calculated tensile strength retention of GFRP bars is summarized in Figure 4” should be modified to “The tensile strength retentions of GFRP bars are summarized in Figure 4” .“The curves of the GFRP bars under stress levels of 0 and 20% were close to each other.” should be modified to “The curves of the GFRP bars under stress levels of 0 and 20% were close to each other as shown in Figure 4”. (line 171,176-177).

Point 12: Table 4 and 5: In the sample designation definition in Lines 143-146, the second number is defined as temperature. Given this designation the first set of samples in each table would have been at 0 deg C which was not described in the Methods.

Response 12: The first number in the sample designation should be defined exposure temperature and the second number defined as the sustained stress level. There is a mistake in the original article.

Point 13: Section 3.2 G1 and G2 GFRP bars are referenced. How are G1 and G2 different from the bars in this study? This difference or similarity is necessary to place the tensile strength retention comparisons in context.

Response 13: I think this suggestion is very important to improve the reliability of the conclusions of the paper. However, the manufacturing process of GFRP bars is not mentioned in the literature, and it is impossible to compare the quality of GFRP bars from the production process. Therefore, the purpose of this section is to verify the applicability of GFRP bars used in this test by comparing the results of previous research.

Point 14: Results and Discussion. There is no comparison of the GFRP bars to conventional iron bars. Such a comparison might help put the results in context.

Response 14: Corrosion of steel reinforcement is a major cause of deterioration of reinforced concrete structures. GFRP bars are used in concrete structures due to their resistance to corrosion. Therefore, it is not necessary to compare the durability of GFRP and steel bars.

Point 15: Figure 8: (a) and (b) panels are not designated. Are these single sample figures (there are no error bars)?

Response 15: These error bars have been added to figure 8 of the revised manuscript.

Point 16: Section 3.5. How were the cross-section samples prepared? This information should be included in the Methods section.

Response 16: This sentence “To prepare the samples, the GFRP bar were first embedded in epoxy resin, cut with a low-speed saw, and then polished with sandpaper and fluffy cloth.” is added in section 3.5 of the revised manuscript to describe the sample making process.(line 157-159)

Point 17: Equation (4) is described as the “logarithm of time” where it is actually the logarithm of the rate constant.

Response 17: It has been shown in Formula 3 that k is the degradation rate (1/time). Therefore, Equation (4) is described as the “logarithm of time”.

Point 18: The logic of the Equation (5) description to the last sentence in Section 4.2 is unclear as to the connection to Equation 2.

Response 18: Equation (2) can only predict the long-term durability of GFRP bars at a given temperature. If the transformation of degradation rate between different temperatures is realized by Equation (5), then Equation (2) can predict the long-term durability of GFRP Bars at different temperatures.

Point 19: Section 4.3 again refers to the “logarithm of time” with respect to Equation

Response 19: Refer to answer 17.

Reviewer 3 Report

the article is well written scientifically.

it needs more English editing. especially in materials and methods section. 

Please explain the epoxy used more in details. the type of epoxy the producer and the properties of the epoxy.  

Figure 1 resolution is not good please revise. 

please explain why only two strains were chosen? any design code was followed for this? please explain the parameters chosen for the study in more details, it is not clear why in section 2.2. 

please choose a better text color for figure 2. 

Can you explain why no international standard was followed for the tests?

Please revise figure 3, it is good to make it into figure 3a and figure 3b, and come up with different legends. Also it is low resolution, please revise and use brighter picture. 

please discuss the results from the comparison with ACI more in details, please refer to more literature to justify, it lacks the strength for justification and discussion. The methods are well described but the discussion needs to be written more in details.

The prediction method is very interesting , perhaps reference to some other works with the same model used could be helpful for the readers.

Author Response

We greatly appreciate you for reviewing our manuscript. Each comment will be very helpful for us to improve our writing skills. All issues have been answered in the responses and modified in the revised manuscript according to your comments.

Point 1: Please explain the epoxy used more in details. the type of epoxy the producer and the properties of the epoxy.

Response 1: Table 2 lists the types of epoxy resin (Bisphenol-A E44). The manufacturer of GFRP bars did not provide the manufacturer and specific parameters of epoxy resin. After our inquiry, the manufacturer of GFRP bars has not yet provided the specific manufacturer of epoxy resin.

Point 2: Figure 1 resolution is not good please revise.

Response 2: Figure 1 has been replaced with a high-resolution picture in the revised manuscript.

Point 3: please explain why only two strains were chosen? any design code was followed for this? please explain the parameters chosen for the study in more details, it is not clear why in section 2.2.

Response 3: These levels of strains are about 1.68–3.35 times the values recommended by ACI 440.1R-15 for creep rupture strain (Table.2). This was done to explore the material’s potentials and evaluate how conservative the current codes and guidelines are. This expression has been added to the revised manuscript. (line 103-106).

Point 4: please choose a better text color for figure 2.

Response 4: The text color of Figure 2 has been modified in the revised manuscript.

Point 5: Can you explain why no international standard was followed for the tests?

Response 5: The compositions of the two simulated solutions is shown in Table 3, and the pH of the two solutions meets the requirements of ACI 440.3R-04(line 108-109) and the specimen was subjected to the tensile test according to the ACI 440.3R B.2-04(line 140-141). Can you tell me in detail that the test did not follow international regulations?

Point 6: Please revise figure 3, it is good to make it into figure 3a and figure 3b, and come up with different legends. Also it is low resolution, please revise and use brighter picture.

Response 6:The legend is given and Figure 3 is replaced with a high-resolution picture. in the revised manuscript.

Point 7: please discuss the results from the comparison with ACI more in details, please refer to more literature to justify, it lacks the strength for justification and discussion. The methods are well described but the discussion needs to be written more in details.

Response 7:References 6 and 16 are added to the revised manuscript to discuss the creep design strain. In addition, references 3, 4 and 24 are added to discuss the design strength

Point 8: The prediction method is very interesting , perhaps reference to some other works with the same model used could be helpful for the readers.

Response 8:References [4, 7, 18, 20, 21, 28,] apply the same model as this paper to predict the long-term durability of GFRP bars. In this paper, the activation energy (Ea) is used to compare the research results in this paper with those in the literature to prove the reliability of the prediction.